# The Desotamide Family of Antibiotics

**DOI:** 10.3390/antibiotics9080452

**Published:** 2020-07-27

**Authors:** Asif Fazal, Michael E. Webb, Ryan F. Seipke

**Affiliations:** 1Astbury Centre for Structural Molecular Biology, Faculty of Biological Sciences, University of Leeds, Leeds LS2 9JT, UK; cm12a2f@leeds.ac.uk; 2Astbury Centre for Structural Molecular Biology, School of Chemistry, University of Leeds, Leeds LS2 9JT, UK; m.e.webb@leeds.ac.uk

**Keywords:** natural products, non-ribosomal peptides, NRPS, standalone enzymes, cyclases

## Abstract

Microbial natural products underpin the majority of antimicrobial compounds in clinical use and the discovery of new effective antibacterial treatments is urgently required to combat growing antimicrobial resistance. Non-ribosomal peptides are a major class of natural products to which many notable antibiotics belong. Recently, a new family of non-ribosomal peptide antibiotics were discovered—the desotamide family. The desotamide family consists of desotamide, wollamide, surugamide, ulleungmycin and noursamycin/curacomycin, which are cyclic peptides ranging in size between six and ten amino acids in length. Their biosynthesis has attracted significant attention because their highly functionalised scaffolds are cyclised by a recently identified standalone cyclase. Here, we provide a concise review of the desotamide family of antibiotics with an emphasis on their biosynthesis.

## 1. Introduction

The majority of antibiotics in use are derived from, or inspired by, microbial natural products and, in particular, secondary metabolites produced by *Streptomyces* species and other filamentous actinobacteria [1]. *Streptomyces* species have complex lifecycles that begin with spore germination followed by growth of vegetative hyphae and end with the production of reproductive unigenomic spores [2]. The production of aerial hyphae and spores is triggered by stress and is frequently, though not always, accompanied by the production of secondary metabolites [3]. These metabolites are presumably used as chemical weapons against competing organisms and/or as signalling molecules to neighbouring microbes [4].

Non-ribosomal peptides (NRPs) are a well-studied family of natural products. NRPs are structurally complex and diverse compounds, often with biologically or therapeutically important activities. Their biosynthesis, as their name indicates, is independent from the ribosome, and is typified by the biosynthetic pathways for gramicidin and tyrocidine, which were amongst the earliest to be studied in detail [5]. NRP biosynthetic systems are composed of large multifunctional enzymes called non-ribosomal peptide synthetases (NRPSs), which are large assembly-line like machines organised into modules whose biochemical function is to incorporate a single monomeric building block until the final polypeptide is generated. Biosynthetic modules are grouped into three categories: Loading modules, elongation modules and termination modules.

A loading module typically consists of two domains: An adenylation (A) domain, which activates an amino acid substrate and loads it onto the second domain, and a peptidyl carrier protein (PCP), which is catalytically inactive, but conformationally dynamic. Elongation modules are also comprised of an A and PCP domain, but possess a condensation (C) domain as well. This precedes the A domain (C-A-PCP) and serves to form a peptide bond between two peptide monomers. Both loading and elongation modules can harbour additional tailoring domains such as epimerase or methyltransferase domains that modify peptide intermediates. During biosynthesis, the growing peptide chain remains covalently linked to the 4′-phosphopantetheinyl cofactor of the PCP domain, presumably to avoid dissolution of the growing peptide chain and to ensure the correct peptide sequence is formed. The terminal biosynthetic module usually possesses a C-terminal thioesterase (TE) domain, which offloads the polypeptide intermediate from the PCP domain onto a conserved serine residue, after which either a hydrolytic or macrocylisation reaction occurs to produce the mature peptide or depsipeptide. Thioesterase domains belong to the large and relatively diverse group of α/β hydrolases and show varying levels of substrate selectivity; when present within an NRPS system it is explicitly required for production of the NRP. Several alternative release domains have also been discovered and characterised at the termini of NRP biosynthetic systems, such as reductase (R) and specialised condensation domains [6,7], as well as rare occurrences of spontaneous offloading or cyclisation [8,9,10]. These offloading domains are *cis*-encoded within the terminal biosynthetic module; however, a novel standalone offloading/cyclase enzyme belonging to the β-lactamase superfamily was recently characterised for the desotamide family of antibiotics, whose biosynthesis are the focus of this review [11,12,13].

## 2. Members of the Desotamide Family of Antibiotics and Their Bioactivities

Compounds within the desotamide family of cyclic peptide antibiotics (Figure 1) range in size between six and ten amino acids in length and are cyclised by a standalone cyclase enzyme belonging to the β-lactamase superfamily. They are typified by the presence of at least one tryptophan or phenylalanine residue and a C-terminal glycine or d-amino acid (which is a prerequisite for cyclisation of the peptide (elaborated upon in Section 5)); they also frequently contain modified or unusual amino acids (elaborated upon in Section 4). The founding member of the desotamide family is desotamide A, which was originally discovered in 1997 from the fermentation broth of *Streptomyces* sp. NRRL 21611 [14]. Six years later, a suite of structurally similar cyclic octapeptides named surugamides A-E were discovered from a marine microbe named *Streptomyces* sp. JAMM992 [15] followed by four additional desotamide analogues (desotamides B-D) produced by *S. scopuliridis* SCSIO ZJ46 [16]. In the same year, a further two desotamide analogues (E and F) were discovered as well as C-terminal d-ornithine-containing wollamides A and B, produced by the same organism *Streptomyces* nov. sp. MST-115088. Soon after, came the discovery of surugamide F, a linear decapeptide previously unobserved during the initial discovery of surugamides A-E [17]. Additional chlorinated hexapeptide members of the desotamide family were recently identified, including the ulleungmycins produced by *Streptomyces* sp. KCB13F003, noursamycins produced by *S. noursei* ATCC 11455 [18,19], and curacomycin and dechlorocuracomycin produced by *S. curacoi* NBRC 12761 [20].

The antimicrobial target or targets for members of the desotamide family remain unknown, but all compounds possess a minimum inhibitory concentration against Gram-positive indicator organisms that is in the micromolar range and, excitingly, wollamides are active against *Mycobacterium tuberculosis* [21]. A structure–activity relationship study was recently performed with synthetic derivatives of wollamide B, which revealed that the Trp and Leu residues in the first and second positions of the macrocycle, respectively, are essential for bioactivity and that it could be enhanced by altering the C-terminal d-Orn residue to d-Arg or d-Lys, but not to their l-stereoisomers [22].

## 3. Biosynthetic Gene Clusters of the Desotamide Family

Increased access to relatively inexpensive genome sequencing technology has led to a predictable increase in the number of biosynthetic gene clusters to which products have been assigned. BGCs for desotamide, surugamide, ulleungmycin, noursamycin, or curacomycin have been identified [17,18,19,20,23]. The composition and architecture of the four BGCs is typical for NRPS systems in that they possess genes encoding the large modular assembly line that specifies the core peptide scaffold using canonical NRPS biosynthesis logic. For example, the co-linearity principle is obeyed and the number of biosynthetic modules encoded is equal to the number of monomers comprising the mature compound, with the exception of the *sur* BGC, which harbours a total of 18 modules and encodes the production of compounds with two different ring sizes, surugamides A-E (octapeptides) and a linear decapeptide named surugamide F (Figure 2; Figure 3). Additionally, the location of epimerase tailoring domains within the assembly line is consistent with the final stereochemistry of the structurally characterised compound.

As is typical for other NRPS BGCs, members of the desotamide family also encode genes for transcription factors, transport and production of BGC-encoded precursors (the latter is discussed in Section 4). Although these BGCs are clearly expressed under the growth conditions used during their initial characterisation, on the whole there is little insight into their regulation. The *dsa* BGC harbours three transcription factor genes encoding a winged helix-turn-helix DNA binding protein (DsaA) and a canonical two-component system (DsaMN). Deletion of either one of these regulatory genes abolished the production of desotamide [24]. The DsaMN two-component system is also encoded within the *ulm* and *nsm/cur* BGCs and, by extension, it is likely to be essential for production of their respective compounds. Interestingly, this two-component system is not present within the *sur* BGC, but a GntR-family regulator (encoded by *surR*) was recently identified as a repressor of surugamide production and its own expression could be modulated by supplementing growth media with ivermectin [25]. Although there is at least one transport system encoded within each BGC, compound export has only been examined for desotamide, where the genes *dsaKL* encode a classic ABC transporter system that when deleted decreased the titre of desotamide [24].

## 4. BGC-Encoded Precursors and Modified Amino Acids

### 4.1. Ornithine

Whilst not uncommon within NRPS biosynthetic systems, Orn is a non-proteinogenic amino acid with obvious structural similarities to Lys, lacking a single methylene unit in the amine-terminating side chain. The α-amino acid is primarily observed as an intermediate in primary metabolic pathways, such as the urea cycle or as a biosynthetic intermediate to the amino acid Arg, which is largely thought to be the origin of Orn for most Orn-containing NRPs [26]. However, some BGCs encode an amidinotransferase that converts Arg to Orn, for example in gobichelin biosynthesis or biosynthesis of the cyanobacterial toxin, cylindrospermopsin, where the enzyme resembles commonly observed arginine:glycine amidinotransferases [27,28,29]. The *ulm* and *nsm/cur* BGCs (where Orn is the fifth amino acid in the hexapeptide, Figure 3) harbour an orthologue of such an amidinotransferase, which presumably provides a supply of Orn that is spatiotemporally consistent with the needs of the assembly line.

### 4.2. Kynurenine and N-Formyl-l-Kynurenine

*N*-formyl-l-kynurenine (NFK) and kynurenine (Kyn) are observed in desotamides C and D at the first position of the hexacyclic ring, respectively (Figure 1). NFK and Kyn are α-amino acids formed primarily through the decomposition and metabolism of the natural, aromatic amino acid Trp [30]. For instance, the enzyme tryptophan 2,3-dioxygenase oxygenates the indole ring of tryptophan to produce *N*-formyl-l-kynurenine, which can be enzymatically transformed, or spontaneously hydrolysed, to form Kyn. NFK or Kyn can either be directly incorporated into the final compound (e.g., daptomycin) or be used as an intermediate towards a further derivatized precursor (e.g., 3-formamidosalicylate in antimycin) [31,32]. In many if not most cases, genes encoding these processing steps reside within the BGC, however inspection of the desotamide BGC did not identify gene candidates with the required products, leading to the suggestion that NFK/Kyn in desotamide biosynthesis originates directly from the primary metabolite pool [23].

### 4.3. Allo-Isoleucine and Homoleucine

The non-proteinogenic amino acid l-*allo*-Ile is observed in several of the desotamide family antibiotics, including the desotamides, wollamides, ulleungmycins, noursamycins, and curacomycin. Although generally rare among natural products, a handful of l- or d-*allo*-Ile containing compounds have been identified. The biosynthetic pathway leading to the production of the phytotoxin coronatine in the phytopathogenic bacteria *Pseudomonas syringae*, although not solely NRPS dependent, utilises NRPS biosynthetic logic and l-*allo*-Ile as a precursor to synthesise coronamic acid, an integral component of the phytotoxin [33]. Recently, cadasides A and B, calcium-dependent acidic lipopeptides whose BGC was characterised in a functional metagenomics study, were discovered [34]. The BGC encodes 13 NRPS modules as well as six other operons specifying the regulation, biosynthesis and transport of chemical precursors or the final compound. The sixth module in the system is described as adenylating l-Ile before epimerisation to d-*allo*-Ile and incorporation into the growing polypeptide chain.

The biosynthetic origin of the non-proteinogenic l-*allo*-Ile precursor was originally identified from the desotamide and marfomycin pathways [35]. The two-enzyme system consists of an aminotransferase and an isomerase, whose collective action results in isomerisation at the β-carbon of Ile and occurs initially through covalent linking of the α-amino group to pyridoxal phosphate (PLP), itself covalently linked to a Lys within the active site of the aminotransferase. This is followed by two deprotonations at the α- and β-carbons, catalysed by the aminotransferase and isomerase, respectively. Reprotonation of the β-carbon from the opposite side to initial abstraction of the proton results in the formation of the diastereoisomer, l-*allo*-Ile, upon release of the amino acid from the enzyme active site. This amino acid can then be directly adenylated by A domains within NRPS modules and incorporated into growing NRPs.

Ulleungmycin A and noursamycin C contain d-homoleucine at the third position of the hexapeptide cycle, although the presence of this non-proteinogenic amino acid is not observed across all compounds comprising the ulleungmycins and noursamycins. The homoleucine within these two compounds was ascribed to originate from the lack of specificity of the enzymes constituting branched chain amino acid biosynthesis [18]. However, this seems somewhat unlikely as the BGCs typically harbour their own set of precursor biosynthetic genes for rare precursors (for example, *allo*-isoleucine synthesis, described above) and there are putative leucine metabolic genes within the *ulm* and *nsm/cur* BGCs. The presence of homologated amino acids has also been observed in the NRPS-derived metabolites; echinocandins, pneumocandins, and pahayokolides, with dihydroxylated homotyrosine in the two former compounds and homophenylalanine in the latter, respectively, although little is known about the origins of the homologated amino acids themselves [36,37,38].

### 4.4. 3-amino-2-methylpropionic Acid (AMPA)

The linear decapeptide surugamide F is composed of nine amino acids (five with l-configuration and four with d-configuration) and an unusual β-amino acid named 3-amino-2-methylpropionic acid (AMPA) that is installed by the fifth biosynthetic module (Figure 3). AMPA, also known as β-aminoisobutyric acid, is rare in natural products. Surugamide F was identified to possess an AMPA moiety, which was followed by the discovery of the biosynthetic gene cluster for leualacin B, a derivative of leualacins A and C-G, which are synthesised with the demethylated β-Ala at the corresponding position [39]. Prior to the discovery of these NRPs, AMPA was identified in hybrid NRPS/PKS compounds cryptophycins, fusaristatins and aspergillipeptides [40,41,42,43]. The lipid moiety present in the lipopeptides cadaside A and B (described in the preceding subsection) is attached to the β-amino group of AMPA.

AMPA has been shown to be derived from two different sources: pyrimidine degradation and decarboxylation of 3-methyl aspartate. Catabolism of the pyrimidine base thymine begins with NAD(P)H-dependent reduction of the alkene within the heterocyclic six-membered ring, catalysed by dihydropyrimidine dehydrogenase. This is followed by amide bond hydrolysis and ring opening by a dihydropyrimidinase and the formation of AMPA is achieved through release of ammonia and carbon dioxide by β-ureidopropionase [44]. Genes encoding enzymes with similar putative catalytic activities were observed within the cadaside BGC and were putatively assigned as AMPA biosynthetic genes, namely an aldehyde dehydrogenase, α/β hydrolase, and P450 monooxygenase. AMPA biosynthetic genes can be putatively identified within the *sur* BGC: Aldehyde dehydrogenase and α/β hydrolase genes form part of an operon downstream of the NRPS genes.

The biosynthesis of AMPA within the cryptophycin pathway, however, has been shown to occur through a divergent mechanistic route [45]. The enzyme CrpG was shown to have shared amino acid identity with pyruvoyl-dependent aspartate decarboxylases, such as the characterised *E. coli* enzyme PanD, whose active, catalytic form is generated through internal proteolytic cleavage of the initially expressed proenzyme [46]. CrpG was observed to effectively catalyse decarboxylation at the α-carboxylic acid of 3-methylaspartate, leading to the formation of AMPA. The enzyme activity was diastereoselective regarding the substrate as (2S,3R)-3-methylaspartate was preferred by 3-4 orders of magnitude over other diastereoisomers, as well as l- and d-Asp.

### 4.5. Chlorotryptophan

The presence of a halogen, specifically chlorine, within antibiotics and other natural products is often important for its biological activity. For example, the chlorine atoms present in many glycopeptide antibiotics (GPAs), such as vancomycin and teicoplanin, are known to be important for binding its lipid II target, which is anchored within bacterial cell membranes and required for cell wall biosynthesis [47]. The 5-chloro-tryptophan moiety in noursamycins and curacomycin has been shown to be critical for antimicrobial activity [20] and this is presumably also true for ulleungmycins.

The incorporation of Cl in a natural product typically occurs via chlorination of Tyr or Trp. Initial efforts to dissect the mechanism by which tryptophan was chlorinated centred on the non-NRPS biosynthetic pathways for the antibiotic pyrroindomycin, the antifungal pyrrolnitrin, and the antitumour agent, rebeccamycin [48,49,50]. Pyrroindomycin contains a 5-chloroindole moiety derived from tryptophan, whilst the antifungal and antitumour agents are synthesised from 7-chlorinated indoles. The Trp halogenases from these systems utilise a reduced flavin cofactor, produced by a flavin reductase as part of the two-component halogenase, molecular oxygen and a halide ion (Cl^-^) to generate hypochlorous acid (HOCl). The active site HOCl is captured by a neighbouring Lys residue to generate a stable, covalent N-Cl bond on the ε-NH_2_, which then acts as the chlorinating agent upon entry of Trp into the active site. Structural analysis of the PrnA (pyrrolnitrin) and RebH (rebeccamycin) enzymes highlighted the presence of two distinct binding modules specialised for binding the flavin cofactor and the Trp substrate, respectively [51,52,53]. The binding modules were separated by a 10 Å tunnel through which the generated HOCl travels before capture by the active site Lys and delivery to the bound Trp. This represented a novel chlorination mechanism and the expansion of mechanistic repertoire employed by natural product biosynthetic systems.

Many NRPS BGCs harbour a halogenase, but only a handful have been characterised in any detail. Recently, the Tyr halogenases involved in the biosynthesis of the GPAs balhimycins and teicoplanins were recently shown to only chlorinate PCP-bound Tyr and not free Tyr or other peptide intermediates in vitro; this observation was subsequently verified in vivo using an engineered dipeptide-producing NRPS system [54]. Preferential utilization of PCP-bound substrates over free amino acids was also recently shown for the Pro and Tyr halogenase enzymes from the pyoluteorin and C-1027 biosynthetic systems, respectively [55,56]. Taken together, these studies have collectively demonstrated that NRP halogenases appear to preferentially utilize PCP-bound monomers as substrates over free amino acids, which is likely also to be true for the Trp halogenases present within the desotamide family of antibiotics.

### 4.6. β-Hydroxyasparagine

A hallmark of many NRPS biosynthetic systems is the presence of specific enzymes catalysing varied chemical modifications at the β-carbon of amino acid substrates. These modifications can result in methyl or amino substitutions, or commonly the β-hydroxylation of amino acids, as seen in the ulleungmycins, noursamycins and curacomycin with the presence of β-hydroxyasparagine in the mature compounds. A wide variety of amino acids have been shown to be hydroxylated within natural products, including, but likely not limited to; Arg, Asn and Glu, as well as the unnatural amino acid enduracididine [57]. The hydroxylation of Leu and Ile side chains, as well as Asp, has also been observed in the antibiotics bicyclomycin (a cyclodipeptide) and cinnamycin (a ribosomally-encoded and posttranslationally modified peptide), respectively [58,59,60], showing the widespread involvement of hydroxylases across many diverse natural product synthetic pathways.

Siderophores produced by NRPS biosynthetic pathways very often contain β-hydroxylated Asp and His residues, which function as bidentate chelating groups for coordination of metal ions [61,62]. These hydroxyl groups are primarily added by the activity of standalone factors, a catalytic methodology also employed by numerous other NRPS systems. The phytotoxin syringomycin, produced by *Pseudomonas syringae*, contains an l-*threo*-β-hydroxyaspartyl residue, whilst both isomers of d-β-hydroxyglutamate are observed in the antimicrobial kutznerides [63,64]. These chemical modifications predominantly occur on PCP-appended amino acid substrates and are catalysed by standalone non-haem iron oxygenases. Exceptions to this are usually observed in the synthesis or hydroxylation of non-proteinogenic amino acids, such as the biosynthesis of unnatural capreomycidine by the hydroxylation of free Arg and hydroxylation of the non-natural amino acid l-enduracididine, as discovered in the biosynthetic pathways for the viomycins and mannopeptimycins, respectively [57,65]. These chemical modifications, however, are similarly catalysed by the family of non-haem iron, α-ketoglutarate-dependent oxygenases.

Generally, it seems that non-haem iron, α-ketoglutarate-dependent oxygenases act upon PCP-bound intermediates. The target amino acid is held within the active site alongside the co-substrates and an Fe(II) ion. A series of recombination and decomposition reactions lead to the activation of the β-methylene group, allowing subsequent stereospecific transfer of a generated hydroxyl group. Studies are ongoing to describe the nature and timing of hydroxylation events within the biosynthetic pathways for the desotamide family antibiotics, however bioinformatic analysis and similarity to known enzymes suggests a mechanism of hydroxylation commensurate to that described above. The importance and effect of the hydroxyl group on the bioactivity profile of the compounds is unknown, as is much of the detail regarding overall bioactivity, and future studies should focus on elucidating these mechanisms and the potential for targeted advances in bioactive range and potency.

## 5. Peptide Offloading within the Desotamide Family by a Novel Class of Standalone Cyclases

### 5.1. Discovery and Characterisation of a Standalone Peptide Cyclase from the Sur BGC

As noted above, the octapeptides surugamide A-E and decapeptide surugamide F were originally isolated from *Streptomyces* sp. JAMM992 and later from *S. albidoflavus* strains J1074 and S4. [13,15,66]. A key peculiarity of this biosynthetic system was noted early on—the terminal biosynthetic modules lack a TE domain. Inspection of the surugamide BGC initially revealed two candidates named *surE* and *surF*, which encode a β-lactamase and α/β-hydrolase, respectively. Analysis of ∆*surE* and ∆*surF* knockout strains quickly indicated SurE was the cyclase rather than SurF, which may function as a type II thioesterase or proofreading thioesterase [13]. Figure 4 summarises the functionality of SurE. The SurE cyclase contains a Ser-Lys-Tyr-His active site tetrad characteristic of β-lactamases and can cyclise SNAC-thioester mimics of surugamides A and B [11,12,13]. In vitro cyclisation assays utilising a SNAC-thioester mimic of surugamide F revealed that SurE transformed the linear decapeptide intermediate into a new compound named cyclosurugamide F, which was retrospectively observed in trace amounts in crude chemical extracts from the producing organism [67]. However, in a subsequent study by another group who used the same SNAC-surugamide F substrate only a trace amount of cyclosurugamide F was detected and instead accumulation of surugamide F was observed [11].

The substrate specificity of SurE has been partially characterised in vitro, where it has been shown that conservative changes to amino acid composition and stereochemistry at internal positions within octapeptide SNAC substrates are tolerated. Interestingly, SurE obligately requires a d-amino acid at the C-terminus of the peptide chain for cyclisation, which is corroborated by the presence of a C-terminal epimerase domain within the terminal biosynthetic modules for surugamide A and surugamide F. The similarity of the cyclase with penicillin binding proteins (PBPs), which, amongst other catalytic activities, can recognise and remove d-alanine from peptidoglycan precursors, could be a potential rationale for the C-terminal d-amino acid requirement of SurE. A recent study also demonstrated that SurE requires heterochirality at the termini of its octa- or decapeptide substrates [68]. The heterochirality requirement explains the observation that the assembly line for all desotatmide family antibiotics starts with incorporation of an amino acid in the l-configuration and ends with an amino acid in the d-configuration conferred by a conserved C-terminal E-domain within the terminal biosynthetic module. A structural rationale for this requirement was also proposed; the C-terminal amino acid residue of the substrate is accommodated within a hydrophobic pocket with an Arg residue hydrogen bonding to the substrate carbonyl, which was also suggested to be important in excluding the side chains of those terminating in l-amino acids. In vivo studies also showed that SurE was capable of cyclising and offloading a truncated form of surugamide F, formed by exclusion of the final two modules of SurC. The promiscuity in substrate utilization of SurE makes it an attractive biocatalyst for production of cyclic peptides, ranging from eight to ten amino acids in length.

### 5.2. SurE Cyclases are Widespread

One of the fundamental uniting aspects of the desotamide family antibiotics is the presence of a SurE cyclase and the corresponding lack of other *cis*-acting termination strategies. The identification of SurE as the offloading domain for the surugamides laid the foundation for the retrospective identification of the cyclases within the *dsa*, *ulm*, and *nsm/cur* BGCs [18,19,20,23] where DsaJ remains the only other SurE family cyclase to be experimentally verified [24]. The identification of these SurE orthologues was the impetus for a more detailed bioinformatic analysis addressing the question of how widespread this cyclisation strategy could be. From a curated database of 1421 actinobacterial genomes, organisms harbouring orthologues of the SurABCD NRPSs and SurE were identified. Candidate genomes were then analysed with antiSMASH [13] to identify NRPS biosynthetic systems without a *cis*-encoded TE domain, but that possessed an adjacent SurE homologue. A total of 166 organisms contained at least one NRPS system employing a standalone cyclase offloading strategy. These BGCs were subsequently anlaysed with BiG-SCAPE [69] to generate a BGC similarity network comprising 15 related subnetworks and 12 singletons. The network contained all members of the desotamide family except for the *dsa* BGC itself suggesting a distant relatedness between this BGC and other members of the family. Upon inspection of the network it became apparent that the number of biosynthetic modules in an NRPS system (4–10 modules) was a major factor influencing formation of subnetworks [13], which is intriguing as it suggests SurE and its orthologues may be able to cyclise peptides ranging in length and composition. Excitingly, the mannopeptimycin BGC harbours a SurE cyclase (MppK), suggesting that this method of release/cyclisation may not be restricted to the desotamide family [40].

## 6. Conclusions and Further Perspectives

The desotamide family of antibiotics are relatively newly discovered antibacterial compounds that appear to be widely produced by *Streptomyces* species. Traditional MIC-based assessment of bioactivities indicates the family generally has micromolar potency against Gram-positive organisms and excitingly this includes *Mycobacterium* species for the wollamides. Nothing is known about their mechanism(s) of action and interrogating this question should be a key goal for future work as should ascertaining essential chemical moieties for bioactivity, building off the existing SAR analysis of wollamides. At first glance, the desotamide family look like quintessential NRPs; however, the presence of unusual chemical moieties within the family motivated researchers to look deeper into their biosynthesis. These studies revealed the presence of a surprisingly large number of genes for the production or modification of precursor amino acids, many of which are involved in controlling stereochemistry while others append additional functional moieties. While most questions in this area have been addressed, there are still gaps in knowledge concerning when exactly some events occur, for example chlorination of Trp and hydroxylation of d-Asp. Arguably, the most surprising biosynthetic feature of the desotamide family is the use of a β-lactamase standalone cyclase for chain release and cyclisation. The standalone nature of this cyclase suggests it will be more easily repurposed than *cis*-encoded TE domains for enhancing chemical synthesis of cyclic peptide antibiotics and therapeutics. However, more detailed analysis of substrate utilisation is required and ideally accompanied by structural data to provide a roadmap for re-engineering studies to exploit the cyclase.

## Figures and Tables

**Figure 1 antibiotics-09-00452-f001:**
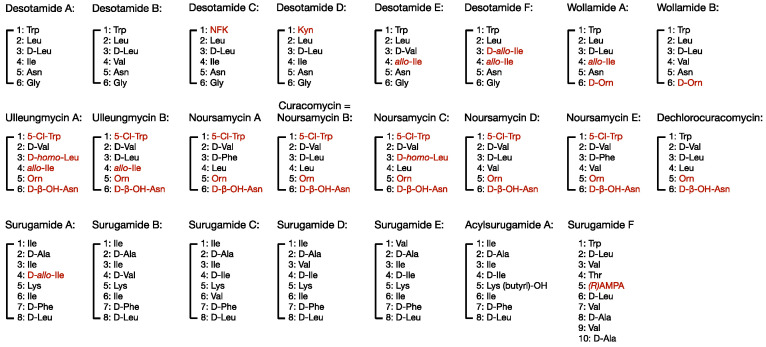
The desotamide family of antibiotics. Numbering indicates the amino acid position within the macrocycle. Red coloured text indicates biosynthetic precursors discussed in Section 4. NFK = *N*-formyl-kynurenine, Kyn = kynurenine, AMPA = 3-amino-2-methylpropionic acid.

**Figure 2 antibiotics-09-00452-f002:**
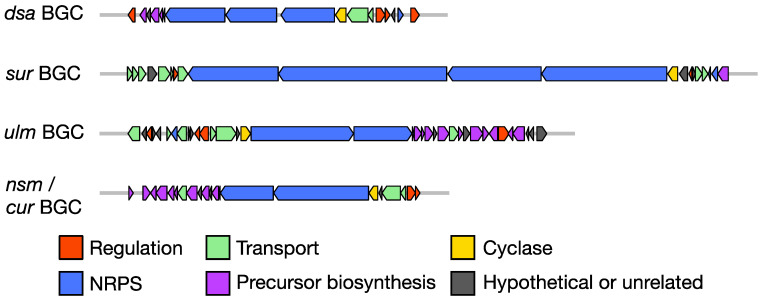
Biosynthetic gene clusters (BGCs) of the desotamide family of antibiotics. BGCs are drawn to relative scale. Colour-coding denotes deduced functionality of gene products. *dsa* = desotamide, *sur* = surugamide, *ulm* = ulleungmycin, *nsm* = noursamycin, *cur* = curacomycin.

**Figure 3 antibiotics-09-00452-f003:**
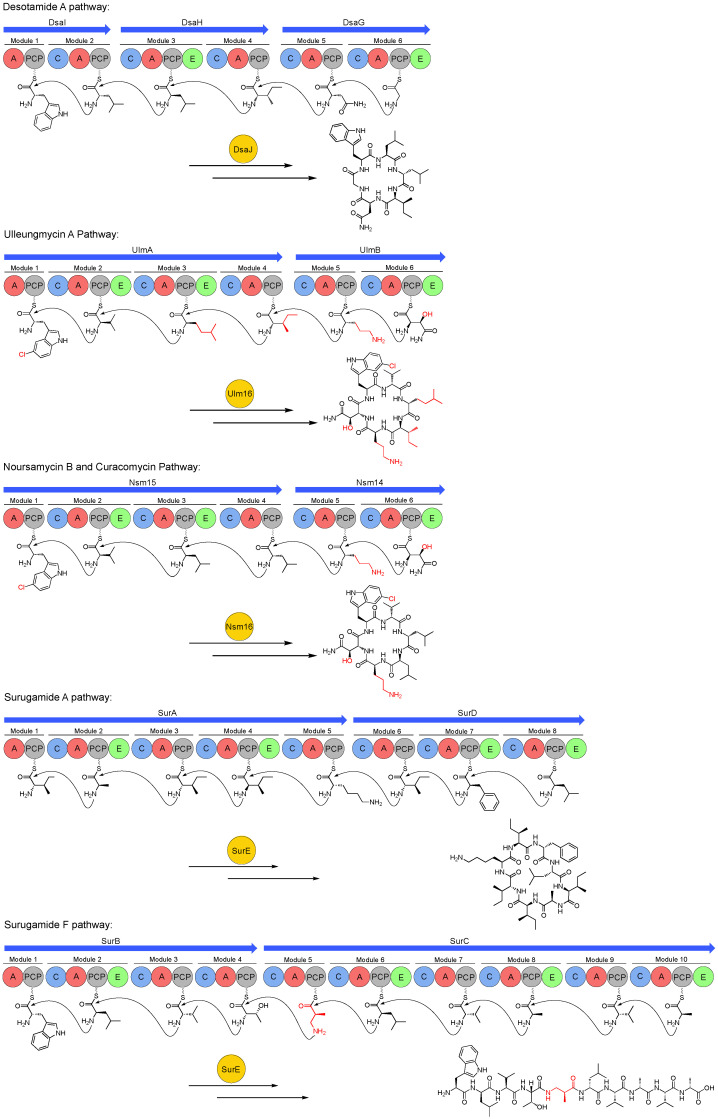
Biosynthetic pathways of the desotamide family of antibiotics. Red colouring denotes biosynthetic precursors discussed in Section 4. A = adenylation domain, PCP = peptidyl carrier protein, C = condensation, E = epimerase. The biosynthesis of desotamide A, ulleungmycin A, noursamycin B or curacomycin, surugamide A and surugamide F is shown.

**Figure 4 antibiotics-09-00452-f004:**
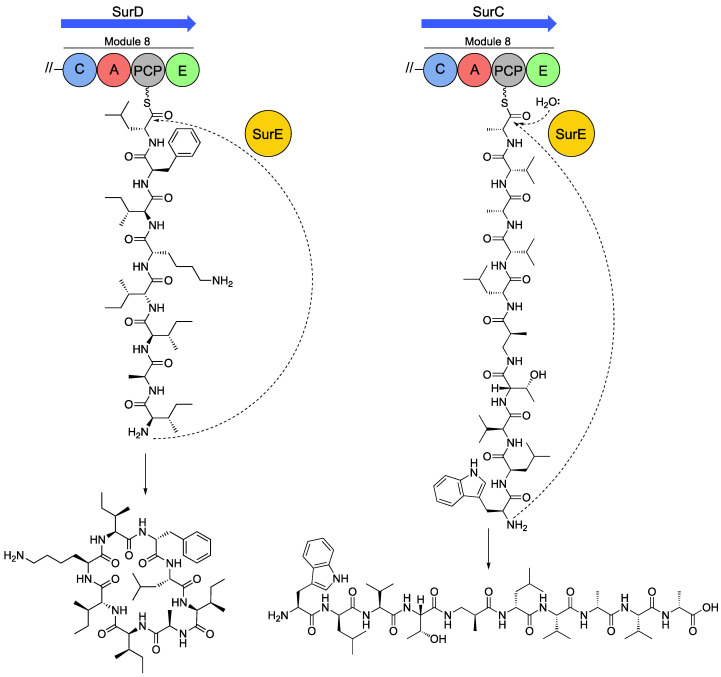
SurE-mediated offloading/cyclisation of surugamide A (**left**) and surugamide F (**right**) from the terminal biosynthetic modules of SurD and SurC, respectively.

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
