# Peer review of "The Desotamide Family of Antibiotics"

_antibiotics, 2020, doi:10.3390/antibiotics9080452_

Round 1

Reviewer 1 Report

A very well written and concise review of a topical group of antibiotic natural products. It will be useful source material. Generally very clear and easy to follow. I personally enjoyed reading it.

Can be accepted as is.

Author Response

Reviewer 1:

A very well written and concise review of a topical group of antibiotic natural products. It will be useful source material. Generally, very clear and easy to follow. I personally enjoyed reading it. Can be accepted as is.

>>Many thanks, indeed.

Reviewer 2 Report

The desotamide family of natural products are a relatively new group of non-ribosomal peptides with potent antimicrobial activities.  The current manuscript comprehensively reviews their chemical structures and biosynthesis, providing a systematic introduction to this interesting family of cyclic peptides. The whole manuscript is well written and in my opinion this paper should be published in Antibiotics. 

1. In figure 3, the biosynthetic pathways were drawn for specific members, not the desotamide, ulleungmycin/noursamycin/curacomycin, and surugamide subgroups of peptides in general. Therefore, the authors should designate the names of specific compounds exemplified in the figure. 

2. The chemical structures of desotamide A and Ulleungmycin A illustrated in figure 3 do not match those from the original literatures, the authors should carefully check the structures of these compounds.

Author Response

Reviewer 2:

The desotamide family of natural products are a relatively new group of non-ribosomal peptides with potent antimicrobial activities.  The current manuscript comprehensively reviews their chemical structures and biosynthesis, providing a systematic introduction to this interesting family of cyclic peptides. The whole manuscript is well written and in my opinion this paper should be published in Antibiotics.

1. In figure 3, the biosynthetic pathways were drawn for specific members, not the desotamide, ulleungmycin/noursamycin/curacomycin, and surugamide subgroups of peptides in general. Therefore, the authors should designate the names of specific compounds exemplified in the figure.

>>Although this is included in the legend, we agree that the in-set headings for each pathway should be more specific and have made this change. Also, see response to the comment below.

2. The chemical structures of desotamide A and Ulleungmycin A illustrated in figure 3 do not match those from the original literatures, the authors should carefully check the structures of these compounds.

>>Many thanks for spotting this. Errors are corrected. We now show the pathways for desotamide A, ulleungmycin A, noursamycin B or curacomycin, suruagmide A and surugamide F. The in-set labels indicate this and it is also specified in the legend. Note, that in the revised manuscript we now include a separate biosynthetic pathway for noursamycin B or curacomycin instead of combining it with ulleungmycin schematic.

Reviewer 3 Report

  1. Page 2, line 51: Change “macrocylisation” to “macrocyclization”

  1. 2, lines 77–78: Change “noursamycins/curacomycins produced . . .” to “noursamycins produced by S. noursei ATCC 11455 [18,19], and curacomycin and dechlorocuracomycin produced by S. curacoi NBRC 12761 [20].”

  1. 2, Figure 1: Change “curacomycin/noursamycin B” to “curacomycin = noursamycin B”

  1. Page 3, line 94: Change “noursamycin/curacomycin” to “noursamycin or curacomycin”

  1. Page 4, line 114: Change “bespoke”. Use a word that is well defined here.

  1. Page 5, line 127: Replace “Bespoke” with a well-defined word or phase.

  1. 5, line 148: Change “(e.g. 3-“ to “(e.g., 3-“

  1. 5, line 154: Change “and noursamycin/curacomycin.” to “noursamycins, and curacomycin.”

  1. Page 7, line 220: Change “noursamycins/curacomycin” to “noursamycins, and curacomycin”

  1. 7, line 251: Change “ulleungmycins and noursamycins/curacomycin” to “ulleungmycins, noursamycins, and curacomycin”

  1. Page 8, line 287: Change “α/β hydrolase” to “α/β-hydrolase”

  1. Page 9, line 307: Change “d-Alanine” to “d-alanine”

Author Response

Reviewer 3:

Page 2, line 51: Change “macrocylisation” to “macrocyclization”

>>We have not made this change. The rest of the document is in “British” English and therefore, in our view, macrocyclisation is correct.

2, lines 77–78: Change “noursamycins/curacomycins produced . . .” to “noursamycins produced by S. noursei ATCC 11455 [18,19], and curacomycin and dechlorocuracomycin produced by S. curacoi NBRC 12761 [20].”

>>Change made.

2, Figure 1: Change “curacomycin/noursamycin B” to “curacomycin = noursamycin B”

>>Change made.

Page 3, line 94: Change “noursamycin/curacomycin” to “noursamycin or curacomycin”

>>Change made.

Page 4, line 114: Change “bespoke”. Use a word that is well defined here.

>>We have changed “bespoke precursors” to “BGC-encoded precursors”.

Page 5, line 127: Replace “Bespoke” with a well-defined word or phase.

>>We have changed “bespoke precursors” to “BGC-encoded precursors”.

5, line 148: Change “(e.g. 3-“ to “(e.g., 3-“

>>Change made.

5, line 154: Change “and noursamycin/curacomycin.” to “noursamycins, and curacomycin.”

>>Change made.

Page 7, line 220: Change “noursamycins/curacomycin” to “noursamycins, and curacomycin”

>>Change made.

7, line 251: Change “ulleungmycins and noursamycins/curacomycin” to “ulleungmycins, noursamycins, and curacomycin”

>>Change made.

Page 8, line 287: Change “α/β hydrolase” to “α/β-hydrolase”

>>Change made.

Page 9, line 307: Change “d-Alanine” to “d-alanine”

>>Change made.

Reviewer 4 Report

The authors wrote a nice review over a group of peptides produced by Streptomycetes.

However, in my opinion the manuscript is a bit biased towards biosynthesis. A better title for the present manuscript might be "Biosynthesis of desotamide-type antibiotics".

The authors would be better off to define which compounds are members of the desotamide family and which are not.
Which are the specific hallmarks that specify a compound being a member?

I did not learn this fact from the manuscript.

Which are related compounds, which do not belong to the family and why?

In addition, the authors should add a paragraph for the bioactivities of the compounds reviewed.

Author Response

Reviewer 4:

The authors wrote a nice review over a group of peptides produced by Streptomycetes.

>>Many thanks, indeed.

However, in my opinion the manuscript is a bit biased towards biosynthesis. A better title for the present manuscript might be "Biosynthesis of desotamide-type antibiotics".

>>This has been discussed amongst the authors. We agree that the majority of the content is dedicated to biosynthesis and have agreed to change the title to that suggested above.

The authors would be better off to define which compounds are members of the desotamide family and which are not. Which are the specific hallmarks that specify a compound being a member? I did not learn this fact from the manuscript. Which are related compounds, which do not belong to the family and why?

>>Most of the requested information is already included in the manuscript and members of the family are shown in Figure 1. In the revised manuscript, we include an additional criterion (they are cyclised by a standalone cyclase enzyme belonging to the β-lactamase superfamily). From the description, and in particular our addendum, we believe it is self-evident why structurally-related compounds are not presently considered to be members of the desotamide family and therefore we have not included further discussion.

In addition, the authors should add a paragraph for the bioactivities of the compounds reviewed.

>>We already included a paragraph describing this (lines 94-101). The bioactivity of these compounds is that they are antibacterial compounds active against Gram-positive bacteria (mechanism of action unknown). We also include a short summary of a SAR analysis conducted with wollamide B.

Round 2

Reviewer 4 Report

The authors addressed my comments more or less appropriately.

Consequently, the paper can be published.